# CONVOLUTIONAL NORMALIZING FLOWS

## ABSTRACT

Bayesian posterior inference is prevalent in various machine learning problems. Variational inference provides one way to approximate the posterior distribution, however its expressive power is limited and so is the accuracy of resulting approximation. Recently, there has a trend of using neural networks to approximate the variational posterior distribution due to the flexibility of neural network architecture. One way to construct flexible variational distribution is to warp a simple density into a complex by normalizing flows, where the resulting density can be analytically evaluated. However, there is a trade-off between the flexibility of normalizing flow and computation cost for efficient transformation. In this paper, we propose a simple yet effective architecture of normalizing flows, *ConvFlow*, based on convolution over the dimensions of random input vector. Experiments on synthetic and real world posterior inference problems demonstrate the effectiveness and efficiency of the proposed method.

## 1 INTRODUCTION

Posterior inference is the key to Bayesian modeling, where we are interested to see how our belief over the variables of interest change after observing a set of data points. Predictions can also benefit from Bayesian modeling as every prediction will be equipped with confidence intervals representing how sure the prediction is. Compared to the maximum a posterior estimator of the model parameters, which is a point estimator, the posterior distribution provide richer information about the model parameter hence enabling more justified prediction.

Among the various inference algorithms for posterior estimation, variational inference (VI) and Monte Carlo markov chain (MCMC) are the most two wisely used ones. It is well known that MCMC suffers from slow mixing time though asymptotically the samples from the chain will be distributed from the true posterior. VI, on the other hand, facilitates faster inference, since it is optimizing an explicit objective function and convergence can be measured and controlled, and it's been widely used in many Bayesian models, such as Latent Dirichlet Allocation (Blei et al., 2003), etc. However, one drawback of VI is that it makes strong assumption about the shape of the posterior such as the posterior can be decomposed into multiple independent factors. Though faster convergence can be achieved by parameter learning, the approximating accuracy is largely limited.

The above drawbacks stimulates the interest for richer function families to approximate posteriors while maintaining acceptable learning speed. Specifically, neural network is one among such models which has large modeling capacity and endows efficient learning. (Rezende & Mohamed, 2015) proposed normalization flow, where the neural network is set up to learn an invertible transformation from one known distribution, which is easy to sample from, to the true posterior. Model learning is achieved by minimizing the KL divergence between the empirical distribution of the generated samples and the true posterior. After properly trained, the model will generate samples which are close to the true posterior, so that Bayesian predictions are made possible. Other methods based on modeling random variable transformation, but based on different formulations are also explored, including NICE (Dinh et al., 2014), the Inverse Autoregressive Flow (Kingma et al., 2016), and Real NVP (Dinh et al., 2016).

One key component for normalizing flow to work is to compute the determinant of the Jacobian of the transformation, and in order to maintain fast Jacobian computation, either very simple function is used as the transformation, such as the planar flow in (Rezende & Mohamed, 2015), or complex tweaking of the transformation layer is required. Alternatively, in this paper we propose a simple

and yet effective architecture of normalizing flows, based on convolution on the random input vector. Due to the nature of convolution, bi-jective mapping between the input and output vectors can be easily established; meanwhile, efficient computation of the determinant of the convolution Jacobian is achieved linearly. We further propose to incorporate dilated convolution (Yu & Koltun, 2015; Oord et al., 2016a) to model long range interactions among the input dimensions. The resulting convolutional normalizing flow, which we term as *Convolutional Flow (ConvFlow)*, is simple and yet effective in warping simple densities to match complex ones.

The remainder of this paper is organized as follows: We briefly review the principles for normalizing flows in Section 2, and then present our proposed normalizing flow architecture based on convolution in Section 3. Empirical evaluations and analysis on both synthetic and real world data sets are carried out in Section 4, and we conclude this paper in Section 5.

## 2 PRELIMINARIES

### 2.1 TRANSFORMATION OF RANDOM VARIABLES

Given a random variable $\boldsymbol{z} \in \mathbb{R}^d$ with density $p(\boldsymbol{z})$, consider a smooth and invertible function $f : \mathbb{R}^d \rightarrow \mathbb{R}^d$ operated on $\boldsymbol{z}$. Let $\boldsymbol{z}' = f(\boldsymbol{z})$ be the resulting random variable, the density of $\boldsymbol{z}'$ can be evaluated as

$$p(\boldsymbol{z}') = p(z) \left| \det \frac{\partial f^{-1}}{\partial \boldsymbol{z}'} \right| = p(z) \left| \det \frac{\partial f}{\partial \boldsymbol{z}} \right|^{-1} \tag{1}$$

thus

$$\log p(\boldsymbol{z}') = \log p(z) - \log \left| \det \frac{\partial f}{\partial \boldsymbol{z}} \right| \tag{2}$$

### 2.2 NORMALIZING FLOWS

Normalizing flows considers successively transforming $\boldsymbol{z}_0$ with a series of transformations $\{f_1, f_2, ..., f_K\}$ to construct arbitrarily complex densities for $\boldsymbol{z}_K = f_K \circ f_{K-1} \circ ... \circ f_1(\boldsymbol{z}_0)$ as

$$\log p(\boldsymbol{z}_K) = \log p(\boldsymbol{z}_0) - \sum_{k=1}^{K} \log \left| \det \frac{\partial f_k}{\partial \boldsymbol{z}_{k-1}} \right| \tag{3}$$

Hence the complexity lies in computing the determinant of the Jacobian matrix. Without further assumption about $f$, the general complexity for that is $\mathcal{O}(d^3)$ where $d$ is the dimension of $\boldsymbol{z}$. In order to accelerate this, (Rezende & Mohamed, 2015) proposed the following family of transformations that they termed as *planar flow*:

$$f(\boldsymbol{z}) = \boldsymbol{z} + \boldsymbol{u}h(\boldsymbol{w}^\top \boldsymbol{z} + b) \tag{4}$$

where $\boldsymbol{w} \in \mathbb{R}^d, \boldsymbol{u} \in \mathbb{R}^d, b \in \mathbb{R}$ are parameters and $h(\cdot)$ is a univariate non-linear function with derivative $h'(\cdot)$. For this family of transformations, the determinant of the Jacobian matrix can be computed as

$$\det \frac{\partial f}{\partial \boldsymbol{z}} = \det(\boldsymbol{I} + \boldsymbol{u}\psi(\boldsymbol{z})^\top) = 1 + \boldsymbol{u}^\top \psi(\boldsymbol{z}) \tag{5}$$

where $\psi(\boldsymbol{z}) = h'(\boldsymbol{w}^\top \boldsymbol{z} + b)\boldsymbol{w}$. The computation cost of the determinant is hence reduced from $\mathcal{O}(d^3)$ to $\mathcal{O}(d)$.

Applying $f$ to $\boldsymbol{z}$ can be viewed as feeding the input variable $\boldsymbol{z}$ to a neural network with only one single hidden unit followed by a linear output layer which has the same dimension with the input layer. Obviously, because of the bottleneck caused by the single hidden unit, the capacity of the family of transformed density is hence limited.

## 3 A NEW TRANSFORMATION UNIT

In this section, we first propose a general extension to the above mentioned planar normalizing flow, and then propose a restricted version of that, which actually turns out to be convolution over the dimensions of the input random vector.

### 3.1 Normalizing flow with $d$ hidden units

Instead of having a single hidden unit as suggested in planar flow, consider $d$ hidden units in the process. We denote the weights associated with the edges from the input layer to the output layer as $\boldsymbol{W} \in \mathbb{R}^{d \times d}$ and the vector to adjust the magnitude of each dimension of the hidden layer activation as $\boldsymbol{u}$, and the transformation is defined as

$$f(\boldsymbol{z}) = \boldsymbol{u} \odot h(\boldsymbol{W}\boldsymbol{z} + \boldsymbol{b}) \tag{6}$$

where $\odot$ denotes the point-wise multiplication. The Jacobian matrix of this transformation is

$$\frac{\partial f}{\partial \boldsymbol{z}} = \text{diag}(\boldsymbol{u} \odot h'(\boldsymbol{W}\boldsymbol{z} + b))\boldsymbol{W} \tag{7}$$

$$\det \frac{\partial f}{\partial \boldsymbol{z}} = \det[\text{diag}(\boldsymbol{u} \odot h'(\boldsymbol{W}\boldsymbol{z} + b))] \det(\boldsymbol{W}) \tag{8}$$

As $\det(\text{diag}(\boldsymbol{u} \odot h'(\boldsymbol{W}\boldsymbol{z} + \boldsymbol{b})))$ is linear, the complexity of computing the above transformation lies in computing $\det(\boldsymbol{W})$. Essentially the planar flow is restricting $\boldsymbol{W}$ to be a vector of length $d$ instead of matrices, however we can relax that assumption while still maintaining linear complexity of the determinant computation based on a very simple fact that the determinant of a triangle matrix is also just the product of the elements on the diagonal.

### 3.2 Convolutional Flow

Since normalizing flow with a fully connected layer may not be bijective and generally requires $\mathcal{O}(d^3)$ computations for the determinant of the Jacobian even it is, we propose to use 1-d convolution to transform random vectors.

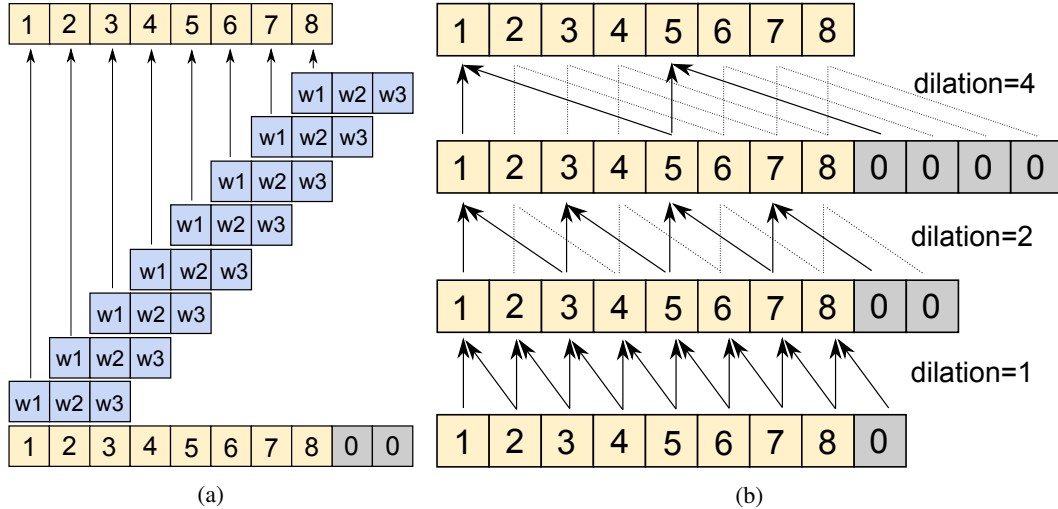

Figure 1: (a) Illustration of 1-D convolution, where the dimensions of the input/output variable are both 8 (the input vector is padded with 0), the width of the convolution filter is 3 and dilation is 1; (b) A block of ConvFlow layers stacked with different dilations.

Figure 1(a) illustrates how 1-d convolution is performed over an input vector and outputs another vector. We propose to perform a 1-d convolution on an input random vector $\boldsymbol{z}$, followed by a non-linearity and necessary post operation after activation to generate an output vector. Specifically,

$$f(\boldsymbol{z}) = \boldsymbol{z} + \boldsymbol{u} \odot h(\text{conv}(\boldsymbol{z}, \boldsymbol{w})) \tag{9}$$

where $\boldsymbol{w} \in \mathbb{R}^k$ is the parameter of the 1-d convolution filter ($k$ is the convolution kernel width), $\text{conv}(\boldsymbol{z}, \boldsymbol{w})$ is the 1d convolution operation as shown in Figure 1(a), $h(\cdot)$ is a monotonic non-linear activation function[1], $\odot$ denotes point-wise multiplication, and $\boldsymbol{u} \in \mathbb{R}^d$ is a vector adjusting

---

[1]Examples of valid $h(x)$ include all conventional activations, including sigmoid, tanh, softplus, rectifier (ReLU), leaky rectifier (Leaky ReLU) and exponential linear unit (ELU).

the magnitude of each dimension of the activation from $h(\cdot)$. We term this normalizing flow as *Convolutional Flow (ConvFlow)*.

ConvFlow enjoys the following properties

- Bi-jectivity can be easily achieved if proper padding and a monotonic activation function are adopted;
- Due to local connectivity, the Jacobian determinant of ConvFlow only takes $\mathcal{O}(d)$ computation independent from convolution kernel width $k$ since

$$\frac{\partial f}{\partial \boldsymbol{z}} = I + \text{diag}(w_1 \boldsymbol{u} \odot h'(\text{conv}(\boldsymbol{z}, \boldsymbol{w}))) \tag{10}$$

  where $w_1$ denotes the first element of $\boldsymbol{w}$.
  For example for the illustration in Figure 1(a), the Jacobian matrix of the 1d convolution $\text{conv}(\boldsymbol{z}, \boldsymbol{w})$ is

$$\frac{\partial \text{conv}(\boldsymbol{z}, \boldsymbol{w})}{\partial \boldsymbol{z}} = \begin{bmatrix} w_1 & w_2 & w_3 & & & & & \\ & w_1 & w_2 & w_3 & & & & \\ & & w_1 & w_2 & w_3 & & & \\ & & & w_1 & w_2 & w_3 & & \\ & & & & w_1 & w_2 & w_3 & \\ & & & & & w_1 & w_2 & w_3 \\ & & & & & & w_1 & w_2 \\ & & & & & & & w_1 \end{bmatrix} \tag{11}$$

  which is a triangular matrix whose determinant can be easily computed;
- ConvFlow is much simpler than previously proposed variants of normalizing flows. The total number of parameters of one ConvFlow layer is only $d + k$ where generally $k < d$, particularly efficient for high dimensional cases. Notice that the number of parameters in the planar flow in (Rezende & Mohamed, 2015) is $2d$ and one layer of Inverse Autoregressive Flow (IAF) (Kingma et al., 2016) and Real NVP (Dinh et al., 2016) require even more parameters. In Section 3.3, we discuss the key differences of ConvFlow from IAF in detail.

A series of $K$ ConvFlows can be stacked to generate complex output densities. Further, since convolutions are only visible to inputs from adjacent dimensions, we propose to incorporate dilated convolution to the flow to accommodate interactions among dimensions with long distance apart. Figure 1(b) presents a block of 3 ConvFlows stacked, with different dilations for each layer. Larger receptive field is achieved without increasing the number of parameters. We term this as a ConvBlock.

From the block of ConvFlow layers presented in Figure 1(b), it is easy to verify that dimension $i \, (1 \le i \le d)$ of the output vector only depends on succeeding dimensions, but not preceding ones. In other words, dimensions with larger indices tend to end up getting little warping compared to the ones with smaller indices. Fortunately, this can be easily resolved by a *Revert Layer*, which simply outputs a reversed version of its input vector. Specifically, a Revert Layer $g$ operates as

$$g(\boldsymbol{z}) := g([z_1, z_2, ..., z_d]^\top) = [z_d, z_{d-1}, ..., z_1]^\top \tag{12}$$

It's easy to verify a Revert Layer is bijective and that the Jacobian of $g$ is a $d \times d$ matrix with 1s on its anti-diagonal and 0 otherwise, thus $\log \left| \det \frac{\partial g}{\partial \boldsymbol{z}} \right|$ is 0. Therefore, we can append a Revert Layer after each ConvBlock to accommodate warping for dimensions with larger indices without additional computation cost for the Jacobian as follows

$$\boldsymbol{z} \to \underbrace{\text{ConvBlock} \to \text{Revert} \to \text{ConvBlock} \to \text{Revert} \to ... \to}_{\text{Repetions of ConvBlock+Revert for } K \text{ times}} f(\boldsymbol{z}) \tag{13}$$

## 3.3 CONNECTION TO INVERSE AUTOREGRESSIVE FLOW

Inspired by the idea of constructing complex tractable densities from simpler ones with bijective transformations, different variants of the original normalizing flow (IAF) (Rezende & Mohamed, 2015) have been proposed. Perhaps the one most related to ConvFlow is Inverse Autogressive

Flow (Kingma et al., 2016), which employs autoregressive transformations over the input dimensions to construct output densities. Specifically, one layer of IAF works as follows

$$f(\boldsymbol{z}) = \boldsymbol{\mu}(\boldsymbol{z}) + \boldsymbol{\sigma}(\boldsymbol{z}) \odot \boldsymbol{z} \tag{14}$$

where

$$[\boldsymbol{\mu}(\boldsymbol{z}), \boldsymbol{\sigma}(\boldsymbol{z})] \leftarrow \text{AutoregreesiveNN}(\boldsymbol{z}) \tag{15}$$

are outputs from an autoregressive neural network over the dimensions of $\boldsymbol{z}$. There are two drawbacks of IAF compared to the proposed ConvFlow:

- The autoregressive neueral network over input dimensions in IAF is represented by a Masked Autoencoder (Germain et al., 2015), which generally requires $\mathcal{O}(d^2)$ parameters per layer, where $d$ is the input dimension, while each layer of ConvFlow is much more parameter efficient, only needing $k + d$ parameters ($k$ is the kernel size of 1d convolution and $k < d$).
- More importantly, due to the coupling of $\boldsymbol{\sigma}(\boldsymbol{z})$ and $\boldsymbol{z}$ in the IAF transformation, in order to make the computation of the overall Jacobian determinant $\det \frac{\partial f}{\partial \boldsymbol{z}}$ linear in $d$, the Jacobian of the autoregressive NN transformation is assumed to be *strictly* triangular (Equivalently, the Jacobian determinants of $\boldsymbol{\mu}$ and $\boldsymbol{\sigma}$ w.r.t $\boldsymbol{z}$ are both always 0. This is achieved by letting the $i$th dimension of $\boldsymbol{\mu}$ and $\boldsymbol{\sigma}$ depend only on dimensions $1, 2, ..., i-1$ of $\boldsymbol{z}$). In other words, *the mappings from $\boldsymbol{z}$ onto $\boldsymbol{\mu}(\boldsymbol{z})$ and $\boldsymbol{\sigma}(\boldsymbol{z})$ via the autogressive NN are always singular, no matter how their parameters are updated, and because of this, $\boldsymbol{\mu}$ and $\boldsymbol{\sigma}$ will only be able to cover a subspace of the input space $\boldsymbol{z}$ belongs to*, which is obviously less desirable for a normalizing flow.[2] Though these sigularity transforms in the autoregressive NN are somewhat mitigated by their final coupling with the input $\boldsymbol{z}$, IAF still performs slightly worse in empirical evaluations than ConvFlow as no singular transform is involved in ConvFlow.

## 4 EXPERIMENTS

We test performance the proposed ConvFlow on two settings, one on synthetic data to infer unnormalized target density and the other on density estimation for hand written digits and characters.

### 4.1 SYNTHETIC DATA

We conduct experiments on using the proposed ConvFlow to approximate an unnormalized target density of $\boldsymbol{z}$ with dimension 2 such that $p(\boldsymbol{z}) \propto \exp(-U(\boldsymbol{z}))$ where $U(\boldsymbol{z}) = \frac{1}{2} \left[ \frac{\boldsymbol{z}_2 - w_1(\boldsymbol{z})}{0.4} \right]^2$ and $w_1(\boldsymbol{z}) = \sin\left(\frac{\pi \boldsymbol{z}_1}{2}\right)$. The target density of $z$ are plotted as the left most column in Figure 2, and we test to see if the proposed ConvFlow can transform a two dimensional standard Gaussian to the target density by minimizing the KL divergence

$$\min KL(q_K(\boldsymbol{z}_k)||p(\boldsymbol{z})) = \mathbb{E}_{\boldsymbol{z}_k} \log q_K(\boldsymbol{z}_k) - \mathbb{E}_{\boldsymbol{z}_k} \log p(\boldsymbol{z}_k) \tag{16}$$

$$= \mathbb{E}_{\boldsymbol{z}_0} \log q_0(\boldsymbol{z}_0)) - \mathbb{E}_{\boldsymbol{z}_0} \log \left| \det \frac{\partial f}{\partial z_0} \right| + \mathbb{E}_{\boldsymbol{z}_0} U(f(\boldsymbol{z}_0)) + \text{const} \tag{17}$$

where all expectations are evaluated with samples taken from $q_0(z_0)$. We use a 2-d standard Gaussian as $q_0(z_0)$ and we test different number of ConvBlocks stacked together in this task. Each ConvBlock in this case consists a ConvFlow layer with kernel size 2, dilation 1 and followed by another ConvFlow layer with kernel size 2, dilation 2. Revert Layer is appended after each ConvBlock, and leaky ReLU with a negative slope of $0.01$ is adopted in ConvFlow.

Experimental results are shown in Figure 2 for different layers of ConvBlock to be stacked to compose $f$. It can be seen that even with 4 layers of ConvBlocks, it's already approximating the target density despite the underestimate about the density around the boundaries. With 8 layers of ConvFlow, the transformation from a standard Gaussian noise vector to the desired target unnormalized density can be accurately learned. Notice that with 8 layers, we are only using 40 parameters ($(4+1) * 8$ with bias terms of convolution counted).

---

[2]Since the singular transformations will only lead to subspace coverage of the resulting variable $\boldsymbol{\mu}$ and $\boldsymbol{\sigma}$, one could try to alleviate the subspace issue by modifying IAF to set both $\boldsymbol{\mu}$ and $\boldsymbol{\sigma}$ as free parameters to be learned, the resulting normalizng flow of which is exactly a version of planar flow as proposed in (Rezende & Mohamed, 2015).

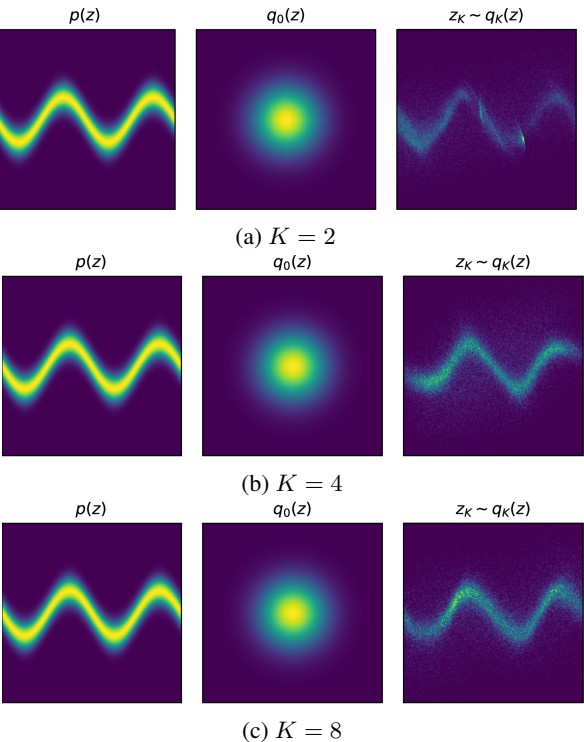

(a) $K = 2$

(b) $K = 4$

(c) $K = 8$

Figure 2: Approximation performance with different number of ConvBlocks

## 4.2 HANDWRITTEN DIGITS AND CHARACTERS

### 4.2.1 SETUPS

To test the proposed ConvFlow for variational inference we use standard benchmark datasets MNIST[3] and OMNIGLOT[4] (Lake et al., 2013). Our method is general and can be applied to any formulation of the generative model $p_\theta(x, z)$; For simplicity and fair comparison, in this paper, we focus on densities defined by stochastic neural networks, i.e., a broad family of flexible probabilistic generative models with its parameters defined by neural networks. Specifically, we consider the following two family of generative models

$$\mathbf{G}_1 : p_\theta(x, z) = p_\theta(z)p_\theta(x|z) \tag{18}$$
$$\mathbf{G}_2 : p_\theta(x, z_1, z_2) = p_\theta(z_1)p_\theta(z_2|z_1)p_\theta(x|z_2) \tag{19}$$

where $p(z)$ and $p(z_1)$ are the priors defined over $z$ and $z_1$ for $G_1$ and $G_2$, respectively. All other conditional densities are specified with their parameters $\theta$ defined by neural networks, therefore ending up with two stochastic neural networks. This network could have any number of layers, however in this paper, we focus on the ones which only have one and two stochastic layers, i.e., $G_1$ and $G_2$, to conduct a fair comparison with previous methods on similar network architectures, such as VAE, IWAE and Normalizing Flows.

We use the same network architectures for both $G_1$ and $G_2$ as in (Burda et al., 2015), specifically shown as follows

$G_1$ : A single Gaussian stochastic layer $z$ with 50 units. In between the latent variable $z$ and observation $x$ there are two deterministic layers, each with 200 units;

---

$G_2$ : Two Gaussian stochastic layers $z_1$ and $z_2$ with 50 and 100 units, respectively. Two deterministic layers with 200 units connect the observation $x$ and latent variable $z_2$, and two deterministic layers with 100 units are in between $z_2$ and $z_1$.

where a Gaussian stochastic layer consists of two fully connected linear layers, with one outputting the mean and the other outputting the logarithm of diagonal covariance. All other deterministic layers are fully connected with tanh nonlinearity. Bernoulli observation models are assumed for both MNIST and OMNIGLOT. For MNIST, we employ the static binarization strategy as in (Larochelle & Murray, 2011) while dynamic binarization is employed for OMNIGLOT.

The inference networks $q(z|x)$ for $G_1$ and $G_2$ have similar architectures to the generative models, with details in (Burda et al., 2015). ConvFlow is hence used to warp the output of the inference network $q(z|x)$, assumed be to Gaussian conditioned on the input $x$, to match complex true posteriors. Our baseline models include VAE (Kingma & Welling, 2013), IWAE (Burda et al., 2015) and Normalizing Flows (Rezende & Mohamed, 2015). Since our propose method involves adding more layers to the inference network, we also include another enhanced version of VAE with more deterministic layers added to its inference network, which we term as VAE+.[5] With the same VAE architectures, we also test the abilities of constructing complex variational posteriors with IAF and ConvFlow, respectively. All models are implemented in PyTorch. Parameters of both the variational distribution and the generative distribution of all models are optimized with Adam (Kingma & Ba, 2014) for 2000 epochs, with a fixed learning rate of 0.0005, exponential decay rates for the 1st and 2nd moments at 0.9 and 0.999, respectively. Batch normalization (Ioffe & Szegedy, 2015) is also used, as it has been shown to improve learning for neural stochastic models (Sønderby et al., 2016).

For inference models with latent variable $z$ of 50 dimensions, a ConvBlock consists of following ConvFlow layers

$$[\text{ConvFlow}(\text{kernel size} = 5, \text{dilation} = 1), \text{ConvFlow}(\text{kernel size} = 5, \text{dilation} = 2),$$
$$\text{ConvFlow}(\text{kernel size} = 5, \text{dilation} = 4), \text{ConvFlow}(\text{kernel size} = 5, \text{dilation} = 8),$$
$$\text{ConvFlow}(\text{kernel size} = 5, \text{dilation} = 16), \text{ConvFlow}(\text{kernel size} = 5, \text{dilation} = 32)] \quad (20)$$

and for inference models with latent variable $z$ of 100 dimensions, a ConvBlock consists of following ConvFlow layers

$$[\text{ConvFlow}(\text{kernel size} = 5, \text{dilation} = 1), \text{ConvFlow}(\text{kernel size} = 5, \text{dilation} = 2),$$
$$\text{ConvFlow}(\text{kernel size} = 5, \text{dilation} = 4), \text{ConvFlow}(\text{kernel size} = 5, \text{dilation} = 8),$$
$$\text{ConvFlow}(\text{kernel size} = 5, \text{dilation} = 16), \text{ConvFlow}(\text{kernel size} = 5, \text{dilation} = 32),$$
$$\text{ConvFlow}(\text{kernel size} = 5, \text{dilation} = 64)] \quad (21)$$

A Revert layer is appended after each ConvBlock and leaky ReLU with a negative slope of $0.01$ is used as the activation function in ConvFlow. For IAF, the autoregressive neural network is implemented as a two layer masked fully connected neural network.

### 4.2.2 GENERATIVE DENSITY ESTIMATION

For MNIST, models are trained and tuned on the 60,000 training and validation images, and estimated log-likelihood on the test set with 5000 importance weighted samples are reported. Table 1 presents the performance of all models, when the generative model is assumed to be from both $G_1$ and $G_2$.

Firstly, VAE+ achieves higher log-likelihood estimates than vanilla VAE due to the added more layers in the inference network, implying that a better posterior approximation is learned (which is still assumed to be a Gaussian). Second, we observe that VAE with ConvFlow achieves much better density estimates than VAE+, which confirms our expectation that warping the variational distribution with convolutional flows enforces the resulting variational posterior to match the true complex posterior. Also, adding more blocks of convolutional flows to the network makes the variational posterior further close to the true posterior. We also observe that VAE with Inver Autoregressive Flows (VAE+IAF) does not always improve likelihood estimates, and even if they do, the improvements are not as significant as ConvFlow. This also confirms our analysis on the singular transformation and

---

[5]VAE+ adds more layers before the stochastic layer of the inference network while the proposed method is add convolutional flow layers after the stochastic layer

Table 1: MNIST test set NLL with generative models $G_1$ and $G_2$ (lower is better $K$ is number of ConvBlocks)

| MNIST (static binarization) | $-\log p(x)$ on $G_1$ | $-\log p(x)$ on $G_2$ |
|---|---|---|
| VAE (Burda et al., 2015) | 87.88 | 85.65 |
| IWAE ($IW = 50$) (Burda et al., 2015) | 86.10 | 84.04 |
| VAE+NF (Rezende & Mohamed, 2015) | - | $\leq 85.10$ |
| VAE+ ($K = 1$) | 87.56 | 85.53 |
| VAE+ ($K = 4$) | 87.40 | 85.23 |
| VAE+ ($K = 8$) | 87.28 | 85.07 |
| VAE+IAF ($K = 1$) | 88.50 | 86.00 |
| VAE+IAF ($K = 2$) | 88.27 | 85.86 |
| VAE+IAF ($K = 4$) | 88.03 | 85.95 |
| VAE+IAF ($K = 8$) | 87.97 | 85.50 |
| VAE+ConvFlow ($K = 1$) | 86.91 | 85.45 |
| VAE+ConvFlow ($K = 2$) | 86.40 | 85.37 |
| VAE+ConvFlow ($K = 4$) | 84.78 | 81.64 |
| VAE+ConvFlow ($K = 8$) | 83.89 | 81.21 |
| IWAE+ConvFlow ($K = 8, IW = 50$) | 79.78 | 78.51 |

subspace issue in IAF. Lastly, combining convolutional normalizing flows with multiple importance weighted samples, as shown in last row of Table 1, further improvement on the test set log-likelihood is achieved. Overall, the method combining ConvFlow and importance weighted samples achieves best NLL on both settings, outperforming IWAE significantly by about 6.3 nats on $G_1$ and 5.5 nats on $G_2$. Notice that, ConvFlow combined with IWAE achieves an NLL of 78.51, slightly better than the best published result of 79.10, achieved by PixelRNN (Oord et al., 2016b). Also it's 1 nat better than the best IAF result of 79.88 reported in (Kingma et al., 2016), which demonstrates the representative power of ConvFlow compared to IAF[6].

Results on OMNIGLOT are presented in Table 2 where similar trends can be observed as on MNIST. One observation different from MNIST is that, the gain from IWAE+ConvFlow over IWAE is not as large as it is on MNIST, which could be explained by the fact that OMNIGLOT is a more difficult set compared to MNIST, as there are 1600 different types of symbols in the dataset. Again on OMNIGLOT we observe IAF with VAE doesn't perform as well as ConvFlow.

### 4.2.3 GENERATED SAMPLES

After the models are trained, generative samples can be obtained by feeding $z \sim N(0, I)$ to the learned generative model $G_1$ (or $z_2 \sim N(0, I)$ to $G_2$). Since higher log-likelihood estimates are obtained on $G_2$, Figure 3 shows the random generative samples from our proposed method trained with $G_2$ on both MNIST and Ominiglot, compared to real samples from the training sets. We observe the generated samples are visually consistent with the training data.

## 5 CONCLUSIONS

This paper presents a simple and yet effective architecture to compose normalizing flows based on convolution on the input vectors. ConvFlow takes advantage of the effective computation of convolution, as well as maintaining as few parameters as possible. To further accommodate long range interactions among the dimensions, dilated convolution is incorporated to the framework without

---

[6]The result in (Kingma et al., 2016) are not directly comparable, as their results are achieved with a more sophiscated VAE architecture and a much higher dimension of latent code ($d = 1920$ for the best NLL of 79.88). However, in this paper, we only assume a relatively simple VAE architecture compose of fully connected layers and the dimension of latent codes to be relatively low, 50 or 100, depending on the generative model in VAE. One could expect the performance of IAF to drop further if simpler VAE architecture and latent codes with lower dimensions are used.

Table 2: OMNIGLOT test set NLL with generative models $G_1$ and $G_2$ (lower is better, $K$ is number of ConvBlocks)

| OMNIGLOT | $-\log p(x)$ on $G_1$ | $-\log p(x)$ on $G_2$ |
|---|---|---|
| VAE (Burda et al., 2015) | 108.86 | 107.93 |
| IWAE ($IW = 50$) (Burda et al., 2015) | 104.87 | 103.93 |
| VAE+ ($K = 1$) | 108.80 | 107.89 |
| VAE+ ($K = 4$) | 108.64 | 107.80 |
| VAE+ ($K = 8$) | 108.53 | 107.67 |
| VAE+IAF ($K = 1$) | 109.44 | 108.74 |
| VAE+IAF ($K = 2$) | 109.69 | 108.36 |
| VAE+IAF ($K = 4$) | 109.47 | 107.61 |
| VAE+IAF ($K = 8$) | 109.34 | 107.43 |
| VAE+ConvFlow ($K = 1$) | 107.41 | 106.32 |
| VAE+ConvFlow ($K = 2$) | 107.05 | 105.80 |
| VAE+ConvFlow ($K = 4$) | 106.24 | 104.35 |
| VAE+ConvFlow ($K = 8$) | 105.87 | 103.58 |
| IWAE+ConvFlow ($K = 8, IW = 50$) | 104.21 | 103.02 |

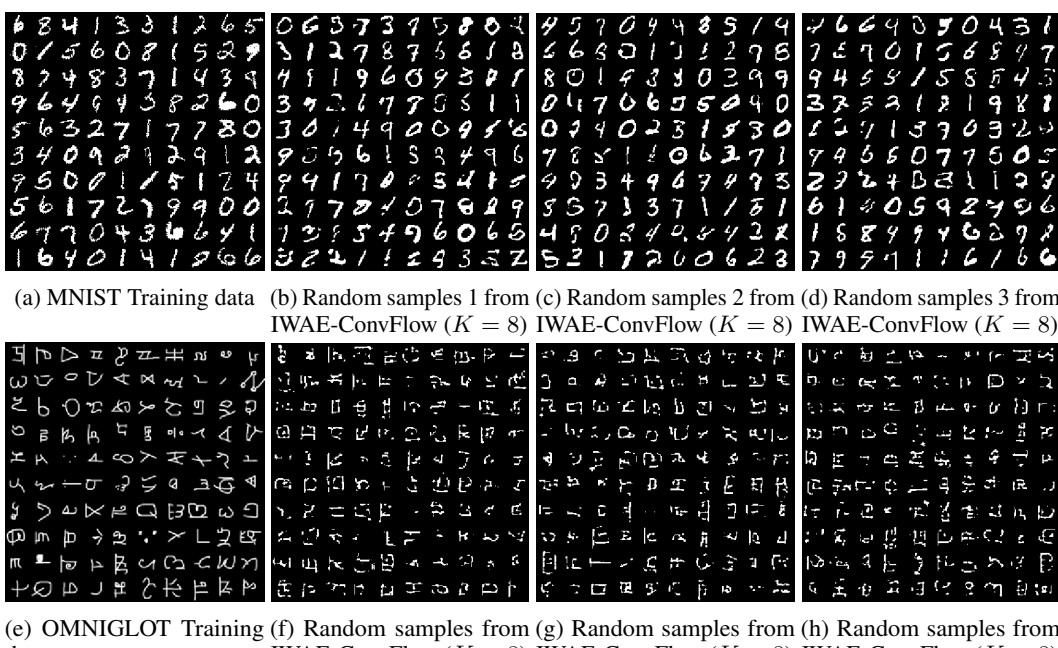

(a) MNIST Training data (b) Random samples 1 from IWAE-ConvFlow ($K = 8$) (c) Random samples 2 from IWAE-ConvFlow ($K = 8$) (d) Random samples 3 from IWAE-ConvFlow ($K = 8$)

(e) OMNIGLOT Training data (f) Random samples from IWAE-ConvFlow ($K = 8$) (g) Random samples from IWAE-ConvFlow ($K = 8$) (h) Random samples from IWAE-ConvFlow ($K = 8$)

Figure 3: Training data and generated samples

increasing model parameters. A Revert Layer is used to maximize the opportunity that all dimensions get as much warping as possible. Experimental results on inferring target complex density and density estimation on generative modeling on real world handwritten digits data demonstrates the strong performance of ConvFlow. Particularly, density estimates on MNIST show significant improvements over state-of-the-art methods, validating the power of ConvFlow in warping multivariate densities. It remains an interesting question as to how many layers of ConvFlows are best to exploit its full performance. We hope to address the theoretical properties of ConvFlow in future work.

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
