# OpenReview forum: "Convolutional Normalizing Flows"
_ICLR.cc/2018/Conference — Reject_

### Official Review · AnonReviewer2 · 2017-11-23
**A form of Inverse Autoregressive Flow?**

**Rating:** 3
**Confidence:** 5

**Review:**

In this paper, the authors propose a type of Normalizing Flows (Rezende and Mohamed, 2015) for Variational Autoencoders (Kingma and Welling, 2014; Rezende et al., 2014) they call Convolutional Normalizing Flows.
More particularly, it aims at extending on the Planar Flow scheme proposed in Rezende and Mohamed (2015). The authors notice an improvement through their method over Normalizing Flows, IWAE with diagonal gaussian approximation, and standard Variational Autoencoders.
As noted by AnonReviewer3, several baselines are missing. But the authors partly address that issue in the comment section for the MNIST dataset.
The requirement of h being bijective seems wrong. For example, if h was a rectifier nonlinearity in the zero-derivative regime, the Jacobian determinant of the ConvFlow would be 1.
More importantly, the main issue is that this paper might need to highlight the fundamental difference between their proposed method and Inverse Autoregressive Flow (Kingma et al., 2016). The proposed connectivity pattern proposed for the convolution in order to make the Jacobian determinant computation is exactly the same as Inverse Autoregressive Flow and the authors seems to be aware of the order dependence of their architecture which is every similar to autoregressive models. This presentation of the paper can be misleading concerning the true innovation in the model trained. Proposing ConvFlow as a type of Inverse Autoregressive Flow would be more accurate and would allow to highlight better the innovation of the work.
Since this work does not offer additional significant insight over Inverse Autoregressive Flow, its value should be on demonstrating the efficiency of the proposed method. MNIST and Omniglot seems insufficient for that purpose given currently published work.
In the current state, I can't recommend the paper for acceptance.


Danilo Jimenez Rezende, Shakir Mohamed: Variational Inference with Normalizing Flows. ICML 2015
Danilo Jimenez Rezende, Shakir Mohamed, Daan Wierstra: Stochastic Back-propagation and Variational Inference in Deep Latent Gaussian Models. ICML 2014
Diederik P. Kingma, Max Welling: Auto-Encoding Variational Bayes. ICLR 2014
Diederik P. Kingma, Tim Salimans, Rafal Józefowicz, Xi Chen, Ilya Sutskever, Max Welling: Improving Variational Autoencoders with Inverse Autoregressive Flow. NIPS 2016

---

> ### Author Response · Authors · 2018-01-05
> **response**
>
> Thank you for your comments and suggestions. Please find our response as follows:
>
> 1. You are right  that the activation function in ConvFlow doesn't have to be  bijective, as there is a skip link from z to the output to account for the bijection even when h returns 0.  Thanks for pointing this out and we have updated our revision accordingly;
>
> 2. We would like to clarify that ConvFlow is not a specific version of IAF, as there are two major differences enjoyed by ConvFlow:
>     a. The number of parameters required for IAF is O(d^2), where d is the input dimension; while ConvFlow only needs k+d, where k is the convolution kernel size, and typically k<d, due to the adoption of 1d convolution
>     b. More importantly, as we shown in the paper revision, the autoregressive NN used in IAF involves singular transformation, thus causing a subspace issue, which effectively limits the representative power for the resulting variable.
>
>    The proposed ConvFlow is able to address the above two drawbacks of IAF and manages to achieve strong results. Please refer to Section 3.3 of the updated paper for a detailed discussion about the differences with IAF.
>
> 3. We have updated with our latest experiments on MNIST and added comparison to IAF based on the same VAE architecture. The latest experiments achieves even slightly better results than thebest published ones, with a best NLL of 78.51 compared to 79.10 achieved by PixelRNN. Also it's  also 1 nat better than the best reported IAF result.
>
>    Please refer to Section 4.2.2 for details about the updated experimental results.

---

### Official Review · AnonReviewer1 · 2017-11-26
**Review of Convolutional Normalizing Flows**

**Rating:** 5
**Confidence:** 4

**Review:**

The paper proposes to increase the expressivity of the variational approximation in VAEs using a new convolutional parameterization of normalizing flows. Starting from the planar flow proposed in Rezende & Mohammed 2015 using a vector inner product followed by a nonliniarity+element-wise scaling the authors suggests to replace inner product with a shifted 1-D convolution. This reduces the number of parameters used from 2*d to k + d and importantly still maintains the linear time computation of the determinant. This approach feels so straightforward that i’m surprised that it have not been tried before. The authors present results on a synthetic task as well as MNIST and OMNIGLOT. Please find some more detailed comments/questions below


Q1) I feel that section 3 could be more detailed about how the convolution normalizing flow relate to normalizing flow, inverse autoregressive flow and the masked-convolution used in real NVP? Especifically a) is it correct that convolutional normalizing flow trades global connectivity for more expressivity locally? b) Can convolutional flow be seen as faster but ´more restricted version of the LSTM implemented inverse autoregressive flow (full lower triangular jacobian vs k off diagonal elements per row in convolutional normalizing flow)

Q2) I miss some more baselines in the experimental section. Did the authors compare the convolutional normalizing flow with e.g. Inverse Autoregressive flow or Auxiliary latent variables?


Q3) Albeit the MNIST results seems convincing - and to a lesser degree the OMNIGLOT ones - I miss results on larger natural image benchmark datasets like cifar10 and ImageNet or preferably other modalities like text? Would it be possible to include results on any of these datasets?

Overall i think the idea is nice and potentially useful due to the ease of implementation and speed of convolutional operations. However I think the authors needs to 1) better describe how their method differs from prior work and 2) compare their method to more baselines for the experiments to fully convincing

---

> ### Author Response · Authors · 2018-01-05
> **response**
>
> Thank you for your comments and suggestions. Please find our response as follows:
>
> 1. Regarding other types of normalizing flows, particularly Inverse Autoregressive Flow (IAF), we would likediscuss two major differences enjoyed by ConvFlow:
>     a. The number of parameters required for IAF is O(d^2), where d is the input dimension; while ConvFlow only needs k+d, where k is the convolution kernel size, and typically k<d, due to the adoption of 1d convolution
>     b. More importantly, as we shown in the paper revision, the autoregressive NN used in IAF involves singular transformation, thus causing a subspace issue, which effectively limits the representative power for the resulting variable.
>
>    The proposed ConvFlow is able to address the above two drawbacks of IAF and manages to achieve strong results. Please refer to Section 3.3 of the updated paper for a detailed discussion about the differences with IAF.
>
> 2. We have updated with our latest experiments on MNIST and added comparison to IAF based on the same VAE architecture. The latest experiments achieves even slightly better results than thebest published ones, with a best NLL of 78.51 compared to 79.10 achieved by PixelRNN. Also it's  also 1 nat better than the best reported IAF result.
>
>    Please refer to Section 4.2.2 for details about the updated experimental results.
>
> 3. Thanks for your suggestions in conducting experiments on larger image datasets, and we have done some preliminary experiments on cifar10. We found out that the simple VAE architecture which achieves strong results on MNIST and OMNIGLOT doesn't give great results on cifar10 compared to PixelRNN, because natural images is much more complicated than MNIST, thus calls for a more sophisticated VAE and we are actively working on that. However, with the simple VAE, we still compare the performance of ConvFlow to IAF on cifar10, and we found out that ConvFlow still gives much better results than IAF. We didn't include the numbers in the paper, as a sophiscated VAE is needed and we plan to add them soon.

---

### Official Review · AnonReviewer3 · 2017-11-27
**novelty is limited. currently no impressive experimental results**

**Rating:** 3
**Confidence:** 4

**Review:**

The authors propose a new method for improving the flexibility of the encoder in VAEs, called ConvFlow. If I understand correctly (please correct me if not) the proposed method is a simplification of Inverse Autoregressive Flow as proposed by Kingma et al. Both of these methods use causal convolution to construct a normalizing flow with tractable Jacobian determinant. The difference is that Kingma et al. used 2d convolution (as well a fully connected architectures) where the authors of this paper propose to use 1d convolution. The novelty therefore seems limited.

The current version of the paper does not present convincing experimental results. The proposed method performs less well than previously proposed methods. If the authors were to update the experimental results to show equal or better performance to SOTA, with an analysis showing their method is indeed computationally less expensive, I would be willing to increase my rating.

---

> ### Author Response · Authors · 2018-01-05
> **response to review**
>
> Thank you for your comments and suggestions. We would like to address your comments as follows:
>
> 1. Regarding IAF, there are two major differences:
>     a. The number of parameters required for IAF is O(d^2), where d is the input dimension; while ConvFlow only needs k+d, where k is the convolution kernel size, and typically k<d;
>     b. More importantly, as we shown in the paper revision, the autoregressive NN used in IAF involves singular transformation, thus causing a subspace issue, which effectively limits the representative power for the resulting variable.
>
>    The proposed ConvFlow is able to address the above two drawbacks of IAF and manages to achieve strong results. Please refer to Section 3.3 of the updated paper for a detailed discussion about the differences with IAF.
>
> 2. We updated with our latest experiments on MNIST and add comparison to IAF based on the same VAE architecture,
>  and our latest experiments achieves slightly better results compared to best published ones, with a best NLL of 78.51 compared to 79.10 achieved by PixelRNN. Also it's 1 nat better than the best IAF result.
>
>    Please refer to Section 4.2.2 for details about updated results.

---

### Comment · AnonReviewer3 · 2017-11-22
**Why no comparison to recent results?**

The conclusion of the paper says "density estimates on MNIST show significant improvements over state-of-the-art methods". This is misleading, as the results table ignores all recent results in this area. E.g. PixelVAE, Lossy VAE, PixelCNN, and IAF-VAE (some of which are cited) all obtain much better results. Is there any reason the proposed method should not be compared against these newer methods?

---

> ### Author Response · Authors · 2017-11-23
> **response**
>
> Thank your for your comment. The results of the methods mentioned in your comment are currently not included in the manuscript, because we would to emphasize that we are NOT putting our focus on optimizing for more sophisticated encoder and decoder architectures as these methods do, but rather on modeling a much richer family of variational posteriors to capture complex distribution of the latent codes on top of a standard encoder network. In fact, we only used  a 2-layer MLP to model the mapping between the input data x to the initial Gaussian latent code z, which is then to be fed into the proposed ConvFlow network to construct complex posteriors. In other words, the proposed method and recent methods, including PixelVAE, PixelCNN, etc, are in orthogonal directions to improve generative modeling and can be potentially combined.
>
> Even with the aforementioned simple encoder and decoder network, our latest experimental results after this submission actually show that we are able to get a NLL of 78.51 with 8 layers of ConvBlock attached on top of the initial Gaussian encoder. This actually surpasses the reported best results of the above methods on statically bainarized MNIST (Deep IAF-VAE: 79.88 and PixelCNN: 79.2) which assumed a much more sophisticated encoder and decoder network (Deep ResNet-like architecture for IAF-VAE and CNN on pixel levels for PixelCNN).
>
> However, we fully agree that providing comparisons to those existing methods helps make the paper a complete story. We will update the new results in the revised paper and release the codes to reproduce the above results.

---

### Author Response · Authors · 2018-01-05
**updated paper revision with comparison to IAF**

1. We added a section detailing the differences of ConvFlow with Inverse Autoregressive Flows (IAF) (Section 3.3);

2. We added updated experimental results on MNIST and OMNIGLOT, as well as comparisons with IAF. (Section 4.2.2)

---

### Decision · Program_Chairs · 2018-01-29
**ICLR 2018 Conference Acceptance Decision**

**Decision:**

Reject

**Comment:**

Thank you for submitting you paper to ICLR. ICLR. Although there revision has improved the paper, the consensus from the reviewers is that this is not quite ready for publication.